# Reconstructing skeletal homeostasis through allogeneic hematopoietic stem cell transplantation in myelofibrosis

Mathias Schäfersküpper[1,4], Alexander Simon [2,3,4], Timur A. Yorgan [2], Felix N. von Brackel [2], Maximilian M. Delsmann[2,3], Anke Baranowsky[2,3], Nico Gagelmann[1], Francis Ayuk [1], Thorsten Schinke [2], Michael Amling[2], Nicolaus Kröger [1,5] ✉ & Tim Rolvien [2,3,5] ✉

Myeloproliferative neoplasm-associated myelofibrosis is a clonal stem cell process characterized by pronounced bone marrow fibrosis associated with extramedullary hematopoiesis and splenomegaly. Allogeneic hematopoietic stem cell transplantation (allo-HSCT) represents the only curative treatment leading to bone marrow fibrosis regression. Here we provide an in-depth skeletal characterization of myelofibrosis patients before and after allo-HSCT utilizing clinical high-resolution imaging, laboratory analyses, and bone biopsy studies. Despite unimpaired bone microarchitecture at peripheral skeletal sites, we observe a marked increase in bone mineral density at the lumbar spine and proximal femur, which is histologically related to severe bone marrow fibrosis and osteosclerosis, fully normalizing after allo-HSCT. Importantly, the regression of fibrosis is accompanied by vanishing osteosclerosis along with restored osteoclastic resorption activity and whole-body calcium homeostasis. Together, our results provide evidence for an extensive reconstruction of skeletal homeostasis by allo-HSCT in MF, leading to rapid resolution of osteosclerosis.

Philadelphia chromosome-negative myeloproliferative neoplasms (MPNs) are hematological disorders of stem cell origin[1]. The group consists of the three entities, primary myelofibrosis (PMF), essential thrombocythemia (ET), and polycythemia vera (PV)[2]. In all entities, the alterations reside within the myeloid compartment[3]. The overwhelming majority of cases show mutational changes affecting the *JAK-STAT* signaling pathway (*CALR, MPL, JAK2*)[4,5]. While the early stages of these diseases differ in clinical and histological features, it remains uncertain whether there is a clear biological distinction between them[6]. Consequently, all three diseases can lead to the same late-stage events[2]. More specifically, there is the possibility of progression to acute myeloid leukemia (AML) and for ET and PV to secondary myelofibrosis (post-ET/PV MF)[6].

PMF or post-ET/PV MF is characterized by advancing extracellular matrix deposition in the bone marrow as well as osteosclerotic changes to the bone itself[7,8]. These changes lead to adaptation by extramedullary hematopoiesis, mainly in the spleen and liver, which can cause symptomatic splenomegaly, portal hypertension, and abdominal pain[9]. Patients in this disease stage often suffer from bone marrow failure leading to cytopenia with increased infection risk, high risk of hemorrhage, and anemia[10]. Inhibition of the *JAK-STAT* signaling pathway may improve overall survival in MF patients but cannot completely stop disease progression[11]. Thus, currently available drugs only result in symptom burden relief, while the only curative option and real disease-modifying modality to date remains allogeneic hematopoietic stem cell transplantation (allo-HSCT)[12]. Because progressive MF is

[1]Department of Stem Cell Transplantation, University Medical Center Hamburg-Eppendorf, Hamburg, Germany. [2]Department of Osteology and Biomechanics, University Medical Center Hamburg-Eppendorf, Hamburg, Germany. [3]Department of Trauma and Orthopaedic Surgery, University Medical Center Hamburg-Eppendorf, Hamburg, Germany. [4]These authors contributed equally: Mathias Schäfersküpper, Alexander Simon. [5]These authors jointly supervised this work: Nicolaus Kröger, Tim Rolvien ✉ e-mail: n.kroeger@uke.de; t.rolvien@uke.de

associated with a limited overall survival, these patients are often found in the transplantation setting. In this regard, it has been outlined that the marrow fibrosis normalizes after successful treatment by allo-HSCT[13,14].

Regarding the specific skeletal changes in MF before and after allo-HSCT, evidence of increased osteoclastogenesis after allo-HSCT has previously been found[13]. Nonetheless, bone biopsies have not yet been analyzed in the course of allo-HSCT on the basis of undecalcified histology and bone-specific histomorphometry, and the respective mechanisms behind MF regression remain unknown. It was previously shown that MPNs pose a higher risk for fractures, although the detailed mechanism behind this observation is not known[15]. Moreover, it is not known whether the transplantation setting poses an additional risk for poor bone quality. An earlier study has shown no major differences in bone radiographic measurements in MF patients compared to controls[16]. Importantly, the previous observation that osteosclerosis in late-stage MF is associated with impaired function of bone-resorptive osteoclasts[17] indicates a close interaction between bone cells and the bone marrow. This is also supported by the fact that MF is characterized by an increased number of monocytes, which represent the osteoclast precursors[18]. In sum, although the bone marrow is encased by cortical envelopes and trabeculae, the contribution of the skeleton (especially bone cells) to MF pathogenesis and regression after allo-HSCT has remained unclear.

In the present study, we outline the longitudinal changes in bone turnover and matrix properties in the setting of curative therapy using allo-HSCT by clinical densitometric and laboratory bone metabolism measurements combined with multiscale iliac crest biopsy analyses. We demonstrate that allo-HSCT leads to a rapid normalization of osteosclerosis, accompanied by restoration of osteoclast function and systemic calcium homeostasis.

## Results

### Study cohort of advanced-stage MF patients

A total of 66 MF patients (34 women, 32 men, $60.3 \pm 8.7$ years) undergoing allo-HSCT were included in this prospective cohort study of whom 40 (61%), 16 (24%), and 10 (15%) had primary MF (PMF), post-ET MF, and post-PV MF, respectively (Supplementary Table 1). Bone marrow fibrosis grade (MF grade)[19] was advanced with 73% of patients having MF-3 and 20% having MF-2. Patients were also found at an advanced stage in prognostic survival models (DIPPS[20] and MYSEC-PM[21]), with 61% classified at intermediate-2 risk and 21% at high risk with a reported median survival time of 4.0 to 4.4 years and 1.5 to 2 years, respectively. There were no statistically significant differences in frequency distributions of clinical characteristics between women and men. Additional demographic characteristics of the study cohort are presented in Supplementary Table 2, sex-specific demographic characteristics are presented in Supplementary Table 3. The propensity score matched healthy control group (n = 66) did not differ in sex distribution, age, weight, height, and body mass index (BMI) (all $p > 0.8$).

### Marked increase in bone mineral density at central sites normalized by allo-HSCT

Osteosclerotic changes in advanced stage MF were visible on dual-energy X-ray absorptiometry (DXA) images of the lumbar spine and hip, which was normalized after allo-HSCT (Fig. 1a). Specifically, we found markedly increased areal bone mineral density (aBMD) values and T-scores ($1.4 \pm 1.8$ vs. $0.0 \pm 1.2$, $p < 0.0001$, d = 0.88) often in the range of osteosclerosis compared to healthy controls at the lumbar spine (Fig. 1b, d). While both were not fully normalized after allo-HSCT in the total cohort ($p < 0.05$, median follow-up 362 days), the analysis of individual patients with available baseline and follow-up measurements (median 377 days) showed a significant decrease in aBMD and T-scores ($1.5 \pm 1.8$ vs. $0.8 \pm 1.9$, $p = 0.009$, d = 0.38). Similar results, but differences with even larger effects, were found at the hip. Namely,

aBMD and T-scores were markedly increased comparing MF patients before allo-HSCT with controls (T-score, $1.4 \pm 1.9$ vs. $-0.6 \pm 0.8$, $p < 0.0001$, d = 1.35). A significant decrease in femoral aBMD and T-score after allo-HSCT was observed in both the total cohort ($p < 0.05$) and individual patients (T-score, $1.8 \pm 2.1$ vs. $0.3 \pm 2.1$, $p < 0.0001$, d = 0.71), leaving no difference when compared to the control group after allo-HSCT ($p > 0.05$) (Fig. 1c, e). Further details are presented in Supplementary Table 2, and consistent results are also evident in the evaluation specific to each sex, as presented in Supplementary Table 3. To further evaluate longitudinal changes in spinal and femoral aBMD, non-linear regression analyses were performed for the percentage change in aBMD based on individual patient courses after allo-HSCT. The best-fit model for spinal aBMD ($R^2 = 0.238$) predicted a change of $-2.6\%$ at 100 days and $-8.0\%$ at 365 days after allo-HSCT (Supplementary Fig. 1a). The best-fit model for femoral aBMD ($R^2 = 0.547$) predicted a change of $-7.2\%$ at 100 days and $-18.4\%$ at 365 days after allo-HSCT in MF (Supplementary Fig. 1b). Of note, when stratified by MF type, patients with secondary MF showed a greater decrease in aBMD T-scores than those with primary MF (Supplementary Fig. 2a, b).

Interestingly, high-resolution peripheral quantitative computed tomography (HR-pQCT) performed at distal radius and distal tibia reference locations showed no osteosclerosis at either location (Fig. 2a). Quantification of radial total volumetric BMD (Tt.vBMD) showed no differences between MF patients before and after allo-HSCT as well as no differences to the control group or in the individual patient trajectories (all $p > 0.5$) (Fig. 2b). There were also no differences between the three groups in other three-dimensional structural parameters such as bone volume to tissue volume (BV/TV, Fig. 2d) or cortical thickness (Ct.Th, Fig. 2f). The same applies to the parameters obtained at the distal tibia (Fig. 2c, e, g). All additional HR-pQCT parameters are presented in Supplementary Table 4. The sex-specific analyses in women (Supplementary Table 5) and men (Supplementary Table 6) confirm these findings.

### Resolution of bone marrow fibrosis is associated with restructuring of the bone matrix

For in-depth histological characterization of iliac crest bone biopsies in the three groups (MF before allo-HSCT, MF after allo-HSCT, and healthy controls), we investigated a total of 60 propensity score matched specimens (20 for each group). The groups were not different with respect to sex distribution, age, BMI (all $p > 0.5$), and serum vitamin D ($p > 0.1$) (Supplementary Table 7). After allo-HSCT we observed a strong regression of marrow fibrosis (Fig. 3a), whereby the bone mass (i.e., osteosclerosis) also normalized (Fig. 3b, c). Specifically, there was a strong decrease in the proportion of marrow area filled with fibrosis tissue after allo-HSCT ($71.0 \pm 25.5\%$ vs. $5.5 \pm 5.9\%$, $p < 0.0001$, d = 3.30), which was only minimally distinguishable from the control group (Fig. 3d). Conversely, bone marrow adipose tissue was significantly increased ($20.5 \pm 21.2\%$ vs. $55.3 \pm 23.2\%$, $p < 0.0001$, d = 1.67), resulting in no detectable difference to the control group (Fig. 3e). No statistically significant differences could be identified regarding the osteoid volume to bone volume (OV/BV), although there was a trend towards a reduction in OV/BV after allo-HSCT (Fig. 3f). This trend was also confirmed statistically when comparing the frequency of osteomalacia (defined as OV/BV > 2%[22]) (Fig. 3g). Furthermore, we found a strong decrease in histologic bone volume fraction ($35.8 \pm 14.9\%$ vs. $16.5 \pm 8.0$, $p < 0.0001$, d = 1.61), mineralized volume fraction, and trabecular number and thickness after allo-HSCT (all $p < 0.05$) (Fig. 3h–k).

### High bone turnover and impaired matrix mineralization persist after allo-HSCT

We found a decrease in bone matrix mineralization in MF patients as measured by quantitative backscattered electron imaging

(qBEI), which persisted after allo-HSCT (Fig. 4a). Specifically, quantification of bone mineral density distribution parameters (Fig. 4b) revealed lower mean calcium content (CaMean) before and after allo-HSCT compared to the control group (Fig. 4c), as well as higher mineralization heterogeneity (CaWidth, Fig. 4d), higher fraction of poorly mineralized matrix (CaLow, Fig. 4e) (all $p \leq 0.001$, $\eta^2 \geq 0.32$), and unchanged fraction of highly mineralized matrix (CaHigh, Fig. 4f) ($p > 0.05$). An increased number of osteocyte lacunae, the mechanosensitive cells embedded in the bone matrix, was partially recovered after allo-HSCT, while the lacunar area was increased both before and after allo-HSCT (all $p < 0.05$) (Fig. 4g, h). This was also reflected by a persistence in high bone formation (i.e., osteoblasts) and resorption (i.e., osteoclasts) parameters after allo-HSCT (all $p < 0.01$, $\eta^2 \geq 0.28$) (Supplementary Fig. 3a-e).

## Evidence of restored osteoclastic resorption capacity and calcium homeostasis after allo-HSCT

Serum parameters of mineral and bone metabolism were available for a larger cohort of 123 MF patients (including 123 before, 120 at 30 days, 112 at 100 days, and 94 at 1 year after allo-HSCT). It was remarkable that the parameters of calcium homeostasis before allo-HSCT showed a consistent picture of low to low-normal calcium levels (Fig. 5a), low 25-hydroxyvitamin D (25-OH-D, Fig. 5c), and elevated parathyroid hormone (PTH, i.e., secondary hyperparathyroidism) (Fig. 5d). In fact, 63% of the patients had secondary hyperparathyroidism, indicating a high calcium demand in the state of osteosclerosis formation. Other parameters such as phosphate, alkaline phosphatase (ALP), bone-specific ALP, osteocalcin, and urinary deoxypyridinoline per creatinine (DPD) were generally not altered before allo-HSCT (Fig. 5b, e–h). There was not only a complete reversal of the disturbed calcium homeostasis

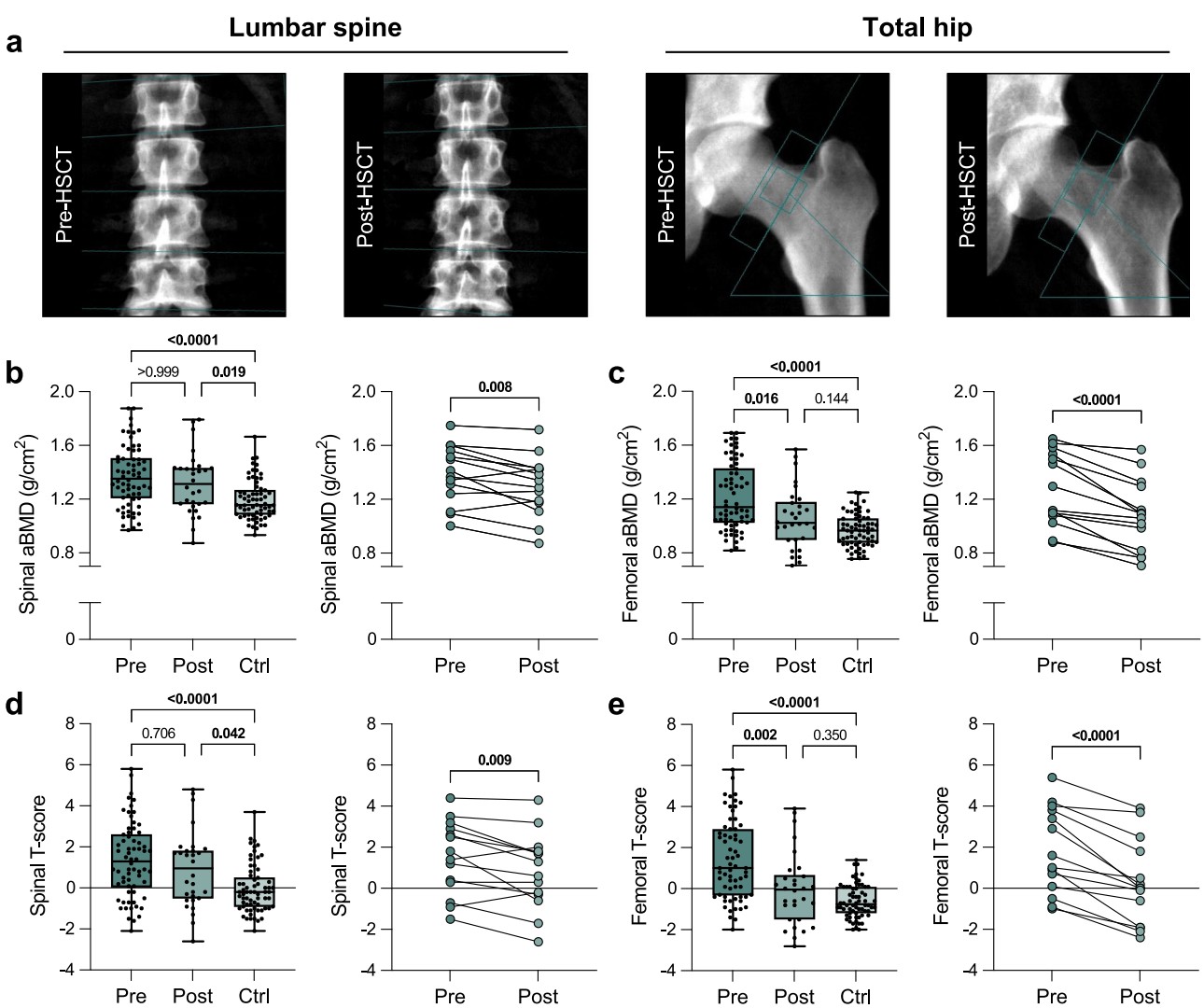

**Fig. 1 | Normalization of increased areal bone mineral density (aBMD) assessed by dual-energy X-ray absorptiometry (DXA) in myelofibrosis after allogeneic hematopoietic stem cell transplantation (allo-HSCT). a** Representative DXA-derived images of the lumbar spine and left hip before (Pre-HSCT) and 1 year after allo-HSCT (Post-HSCT). **b** Spinal aBMD of myelofibrosis patients before allo-HSCT (Pre, n = 66) and after allo-HSCT (Post, n = 30), along with the control group (Ctrl, n = 66). On the right, only patients with individual courses (connected by lines, n = 14) are shown. **c** Femoral aBMD of the three groups (Pre, n = 66; Post, n = 30; Ctrl, n = 66) and individual patient courses (n = 14). **d** Spinal T-score of the three groups (Pre, n = 66; Post, n = 30; Ctrl, n = 66) and individual patient courses (n = 14).

**e** Spinal T-score of the three groups (Pre, n = 66; Post, n = 30; Ctrl, n = 66) and individual patient courses (n = 14). The median (center line, 50th percentile) and interquartile range (box edges, 25th to 75th percentile) with whiskers extending to the minimum and maximum values are depicted. Each data point is shown. Differences between the three groups were calculated using Kruskal-Wallis *H* test with Dunn's test (**b–e**). Differences for the individual courses of patients (Pre and Post, connected points) were calculated using two-tailed paired *t* test (**b–e**). Exact *p*-values of the comparisons are displayed above the brackets and numbers in bold indicate statistical significance (*p* < 0.05). Source data are provided as a Source Data file.

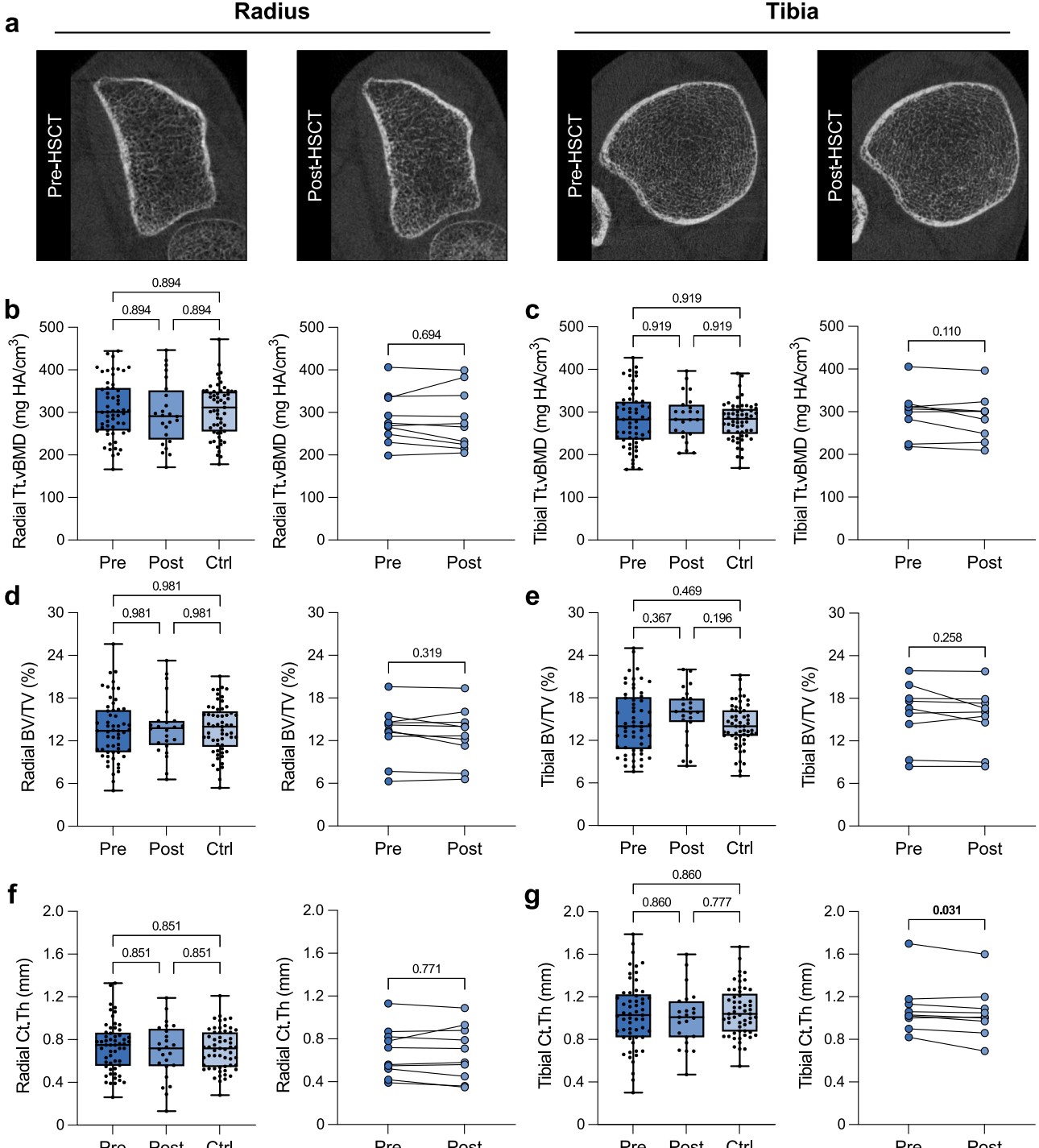

**Fig. 2 | Physiological volumetric bone mineral density (vBMD) and micro-architecture assessed by high-resolution peripheral quantitative computed tomography (HR-pQCT) in myelofibrosis before and after allo-HSCT.**
**a** Representative axial slice of HR-pQCT-derived image of the distal radius and distal tibia before (Pre-HSCT) and one year after allo-HSCT (Post-HSCT). **b** Radial total vBMD (Tt.vBMD) of myelofibrosis patients before allo-HSCT (Pre, n = 57) and after allo-HSCT (Post, n = 24), along with the control group (Ctrl, n = 57). On the right, only patients with individual courses (connected by lines, n = 10) are shown. **c** Tibial Tt.vBMD of the three groups (Pre, n = 57; Post, n = 23; Ctrl, n = 57) and individual patient courses (n = 9). **d** Radial bone volume to tissue volume (BV/TV) of the three groups (Pre, n = 57; Post, n = 24; Ctrl, n = 57) and individual patient courses (n = 10). **e** Tibial BV/TV of the three groups (Pre, n = 57; Post, n = 23; Ctrl, n = 57) and individual patient courses (n = 9). **f** Radial cortical thickness (Ct.Th) of the three

groups (Pre, n = 57; Post, n = 24; Ctrl, n = 57) and individual patient courses (n = 10). **g** Tibial Ct.Th of the three groups (Pre, n = 57; Post, n = 23; Ctrl, n = 57) and individual patient courses (n = 9). The median (center line, 50th percentile) and inter-quartile range (box edges, 25th–75th percentile) with whiskers extending to the minimum and maximum values are depicted. Each data point is shown. Differences between the three groups were calculated using one-way ANOVA with Holm-Šidák test (**b–g**). Differences for the individual courses of patients (Pre and Post, connected points) were calculated using two-tailed paired $t$ test (**b–f**) or Wilcoxon signed-rank test (**g**) for normally and non-normally distributed data, respectively. Exact $p$-values of the comparisons are displayed above the brackets and numbers in bold indicate statistical significance ($p < 0.05$). Source data are provided as a Source Data file.

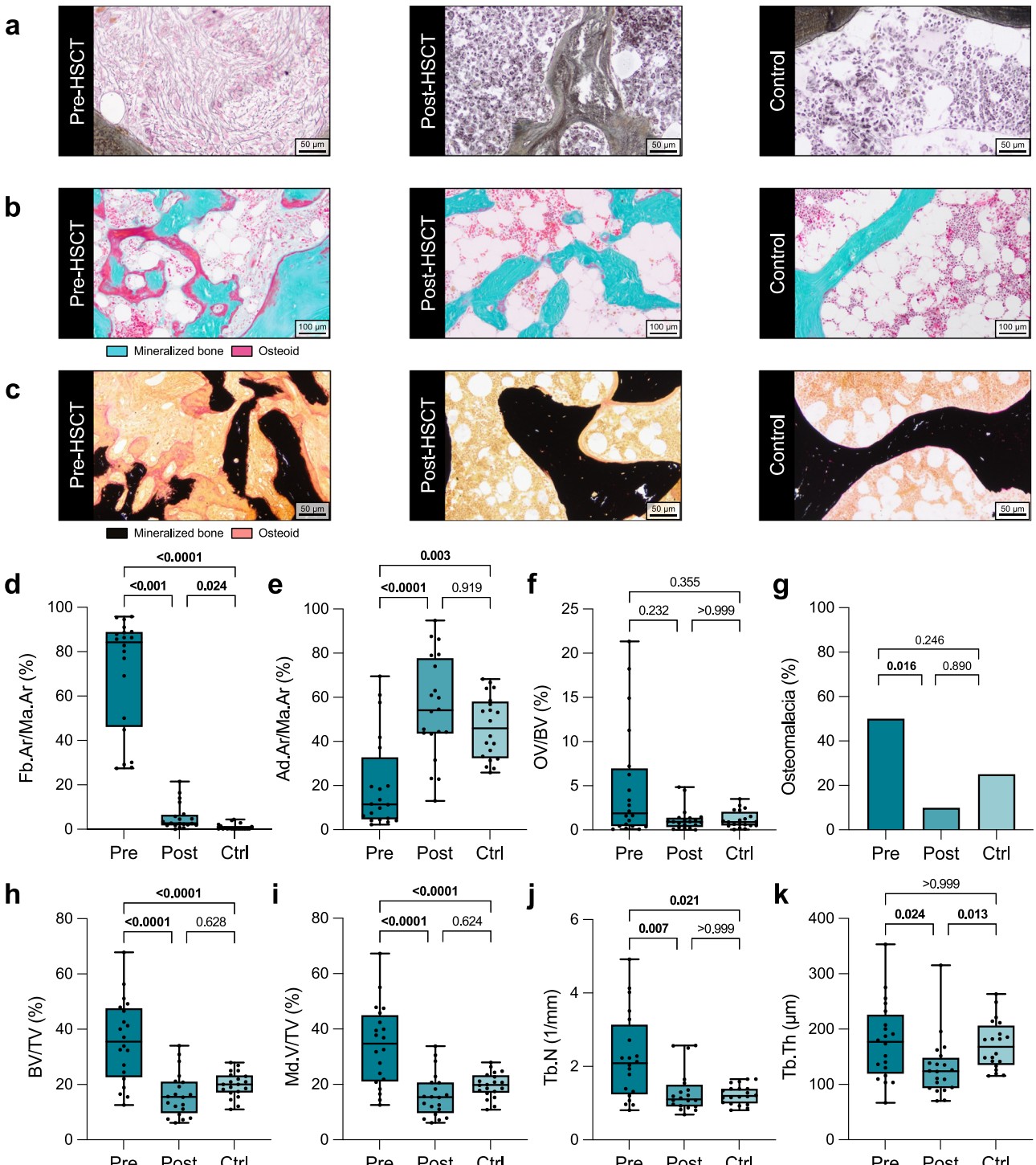

**Fig. 3 | Restoration of histomorphometric bone and bone marrow properties in iliac crest biopsies from myelofibrosis patients after allo-HSCT.**
**a** Representative histological images of Gomori's trichrome-stained sections in myelofibrosis patients before allo-HSCT (Pre-HSCT) and after allo-HSCT (Post-HSCT), along with the control group (Control). **b** Representative histological images of trichrome Masson-Goldner-stained sections of the three groups.
**c** Representative histological images of von Kossa-stained sections of the three groups. **d** Fibrosis area to marrow area (Fb.Ar/Ma.Ar) of myelofibrosis patients before allo-HSCT (Pre, n = 20) and after allo-HSCT (Post, n = 20), along with the control group (Ctrl, n = 20). **e** Adipose area to marrow area (Ad.Ar/Ma.Ar) of the three groups (Pre, n = 20; Post, n = 20; Ctrl, n = 20). **f** Osteoid volume to bone volume (OV/BV) of the three groups (Pre, n = 20; Post, n = 20; Ctrl, n = 20).
**g** Percentage of patients with osteomalacia (defined as OV/BV > 2%[22]) in each group (Pre, n = 20; Post, n = 20; Ctrl, n = 20). **h** Bone volume to tissue volume (BV/TV) of

the three groups (Pre, n = 20; Post, n = 20; Ctrl, n = 20). **i** Mineralized volume to tissue volume (Md.V/TV) of the three groups (Pre, n = 20; Post, n = 20; Ctrl, n = 20). **j** Trabecular number (Tb.N) of the three groups (Pre, n = 20; Post, n = 20; Ctrl, n = 20). **k** Trabecular thickness (Tb.Th) of the three groups (Pre, n = 20; Post, n = 20; Ctrl, n = 20). The median (center line, 50th percentile) and interquartile range (box edges, 25th–75th percentile) with whiskers extending to the minimum and maximum values are depicted. Each data point is shown. For osteomalacia, the percentage of patients in each group is displayed. Differences between the three groups were calculated using one-way ANOVA with Holm-Šidák test (**h**, **i**) or Kruskal-Wallis *H* test with Dunn's test (**d**–**g**, **j**, **k**) for normally and non-normally distributed data, respectively. Exact *p*-values of the comparisons are displayed above the brackets and numbers in bold indicate statistical significance (*p* < 0.05). Source data are provided as a Source Data file.

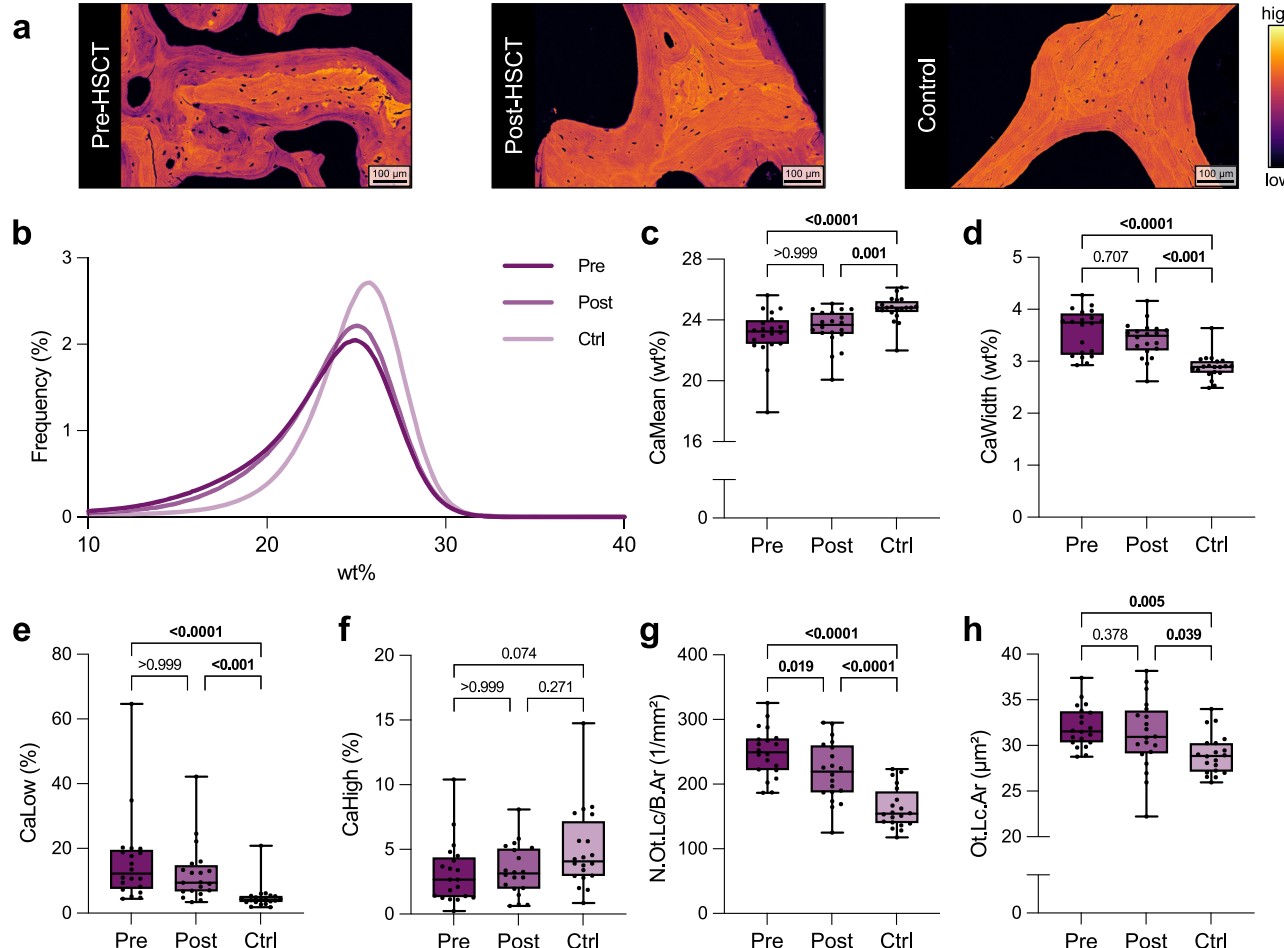

**Fig. 4 | Persistent impairment of bone matrix mineralization and osteocyte lacunar indices in myelofibrosis after allo-HSCT. a** Representative images from qBEI of the three groups. The brightness corresponds to the degree of mineralization (pseudocolors). **b** Bone mineral density distribution (BMDD) curves of the three groups (Pre, n = 20; Post, n = 20; Ctrl, n = 20). **c** Mean calcium content (CaMean) of the three groups (Pre, n = 20; Post, n = 20; Ctrl, n = 20). **d** Mineralization heterogeneity (CaWidth) of the three groups (Pre, n = 20; Post, n = 20; Ctrl, n = 20). **e** Fraction of lowly mineralized matrix (CaLow) of the three groups (Pre, n = 20; Post, n = 20; Ctrl, n = 20). **f** Fraction of highly mineralized matrix (CaHigh) of the three groups (Pre, n = 20; Post, n = 20; Ctrl, n = 20).

**g** Number of osteocyte lacunae per bone area (N.Ot.Lc/B.Ar) of the three groups (Pre, n = 20; Post, n = 20; Ctrl, n = 20). **h** Osteocyte lacunar area (Ot.Lc.Ar) of the three groups (Pre, n = 20; Post, n = 20; Ctrl, n = 20). The median (center line, 50th percentile) and interquartile range (box edges, 25th–75th percentile) with whiskers extending to the minimum and maximum values are depicted. Each data point is shown. Differences between the three groups were calculated using one-way ANOVA with Holm-Šidák test (**g**, **h**) or Kruskal-Wallis *H* test with Dunn's test (**c**–**f**) for normally and non-normally distributed data, respectively. Exact *p*-values of the comparisons are displayed above the brackets and numbers in bold indicate statistical significance (*p* < 0.05). Source data are provided as a Source Data file.

($p < 0.0001$, d = 0.96) with normalization of PTH and 25-OH-D one year after allo-HSCT (both $p < 0.001$, d = 1.53 and d = 1.40), but also a significant increase in the bone resorption marker DPD ($p < 0.0001$, d = 1.33), with a peak value at 100 days after allo-HSCT. Further details are presented in Supplementary Table 8.

**Association of bone mineral density and bone marrow fibrosis**
We detected a significant positive association between MF grade and spinal aBMD (r = 0.324, $p = 0.001$) as well as between MF grade and femoral aBMD (r = 0.543, $p < 0.0001$) (Fig. 6a). However, no significant correlations were found between MF grade and vBMD measured at the distal radius and tibia by HR-pQCT ($p > 0.1$) (Fig. 6b). There were also no significant correlations between fibrosis area to marrow area and either DXA-derived aBMD or HR-pQCT-derived vBMD parameters ($p > 0.1$) (Fig. 6c, d).

## Discussion
In the present study, we demonstrated the distinct changes not only to the bone marrow but also to the bone matrix in patients suffering from MPN-associated MF undergoing allo-HSCT. Specifically, while the

pathologically increased marrow fibrosis and osteosclerosis resolved effectively, parameters of bone turnover and matrix quality were only partly recovered after allo-HSCT. Importantly, however, we found evidence of restored osteoclastic resorption activity leading to restored calcium homeostasis, as indicated by normalized serum calcium and parathyroid hormone levels.

Patients suffering from MPNs display different disturbances of hematopoietic cell differentiation. In MF, increased numbers of monocytes, the precursors of osteoclasts, have been shown to be associated with poorer patient survival[18]. In addition, increased numbers of dysfunctional osteoclasts have previously been reported in MF patients[17]. However, it has been unclear how the regression of marrow fibrosis is linked to changes in osteoclast function after allo-HSCT. While we here demonstrate persistently increased osteoclast numbers in MF before and after allo-HSCT, the increased urinary DPD crosslinks peaking 100 days after allo-HSCT suggest that osteoclasts regain their function, thus contributing to the decline in osteosclerosis. The phenomenon of persistently elevated osteoclast indices may suggest that MF patients are at increased risk of bone loss and osteoporotic fractures in a phase after successful regression of pathological

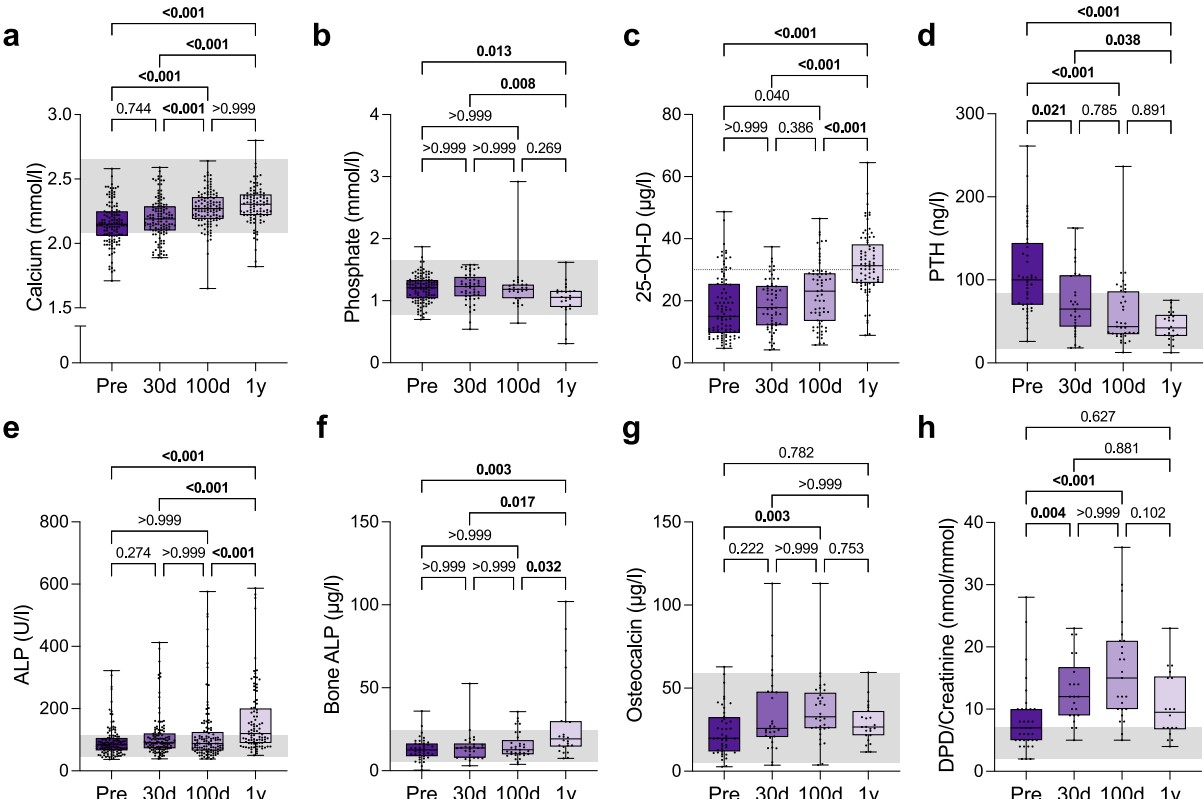

**Fig. 5 | Restoration of calcium homeostasis and bone resorptive activity after allo-HSCT. a** Serum calcium of myelofibrosis patients before allo-HSCT (Pre, n = 123) and at 30 days (30 d, n = 120), 100 days (100 d, n = 112), as well as 1 year (1 y, n = 94) after allo-HSCT. **b** Phosphate of myelofibrosis patients before and after allo-HSCT (Pre, n = 113; 30 d, n = 49; 100 d, n = 28; 1 y, n = 24). **c** 25-hydroxyvitamin D (25-OH-D) of myelofibrosis patients before and after allo-HSCT (Pre, n = 96; 30 d, n = 59; 100 d, n = 63; 1 y, n = 77). **d** Parathyroid hormone (PTH) of myelofibrosis patients before and after allo-HSCT (Pre, n = 43; 30 d, n = 29; 100 d, n = 37; 1 y, n = 24). **e** Alkaline phosphatase (ALP) of myelofibrosis patients before and after allo-HSCT (Pre, n = 123; 30 d, n = 119; 100 d, n = 111; 1 y, n = 92). **f** Bone-specific alkaline phosphatase (Bone ALP) of myelofibrosis patients before and after allo-HSCT (Pre, n = 39; 30 d, n = 29; 100 d, n = 33; 1 y, n = 24). **g** Osteocalcin of myelofibrosis patients

before and after allo-HSCT (Pre, n = 44; 30 d, n = 29, 100 d, n = 33; 1 y, n = 23). **h** Urinary deoxypyridinoline (DPD)/creatinine of myelofibrosis patients before and after allo-HSCT (Pre, n = 35; 30 d, n = 22; 100 d, n = 25; 1 y, n = 18). The median (center line, 50th percentile) and interquartile range (box edges, 25th– 75th percentile) with whiskers extending to the minimum and maximum values are depicted. Each data point is shown. The gray area indicates the reference range. In case of 25-OH-D, the gray line at 30 µg/l marks the threshold value for vitamin D insufficiency. Differences between the groups were calculated using Kruskal-Wallis *H* test with Dunn's test (**a–h**). Exact *p*-values of the comparisons are displayed above the brackets and numbers in bold indicate statistical significance (*p* < 0.05). Source data are provided as a Source Data file.

osteosclerosis by allo-HSCT, underlining the importance of long-term monitoring of fracture risk. Overall, however, the phenomenon of osteosclerosis development and regression in MF remains incompletely understood and further research is needed to elucidate the molecular mechanisms involved. In this context, it is noteworthy that although we found a strikingly increased histologic extent of osteosclerosis and osteoidosis in the bone biopsies before allo-HSCT, the increase in osteoblast numbers was unusually low.

In addition to the observed histological alterations of bone turnover, we were able to outline detailed clinical features in MF patients undergoing allo-HSCT, focusing on BMD and microarchitecture. The high degree of marrow fibrosis and osteosclerosis in the histomorphometric analyses of the iliac crest biopsies clinically translated into increased DXA T-scores. This indicates that DXA might be suitable to estimate the degree of MF in these patients non-invasively. As the grade of fibrosis is associated with poorer prognosis[7,23,24] and fibrosis regression after transplantation predicts patient survival[25], DXA measurements may help in the risk stratification and prognosis evaluation in MF patients. Additional investigation is required to determine if this hypothesis holds true.

Conversely, DXA may be inappropriate in the context of fracture risk assessment. Accurate fracture risk assessment is important because patients suffering from myeloproliferative neoplasms are at

higher risk of osteoporotic fractures[26]. The generally advanced age of MF patients additionally contributes to an increased fracture risk[27], which is likely to increase further with the long-term survival achieved through allo-HSCT. In contrast to DXA, HR-pQCT showed neither increased values before allo-HSCT nor a relevant dynamic in the short-term course after allo-HSCT. Whether this discrepancy is due to the technical differences between DXA and HR-pQCT or to differences in the degree of myelofibrosis and osteosclerosis in central vs. peripheral measurement sites (spine and hip vs. radius and tibia) remains unknown. The latter explanation is supported by the fact that metabolic activity in MF patients has been shown to progress from central to peripheral bone marrow areas, as detected by [18]F-FDG PET/CT[28]. An indirect indication that the decrease in DXA values after allo-HSCT is due to resorbed pathologic bone (i.e., osteosclerosis) rather than actual bone loss in the context of established risk factors such as chemotherapy is the magnitude of BMD reduction compared to other malignancies of the hematopoietic system. Namely, the reduction in aBMD in our cohort, calculated one year after allo-HSCT, was −8.0% at the lumbar spine and −18.4% at the proximal femur (Supplementary Fig. 1a, b), compared to −5.5% at the lumbar spine and −6.8% at the proximal femur in patients with acute lymphoid leukemia (ALL), acute myeloid leukemia (AML), and myeloid dysplastic syndrome (MDS)[29]. Our results suggest that DXA measurements indicate elevated aBMD

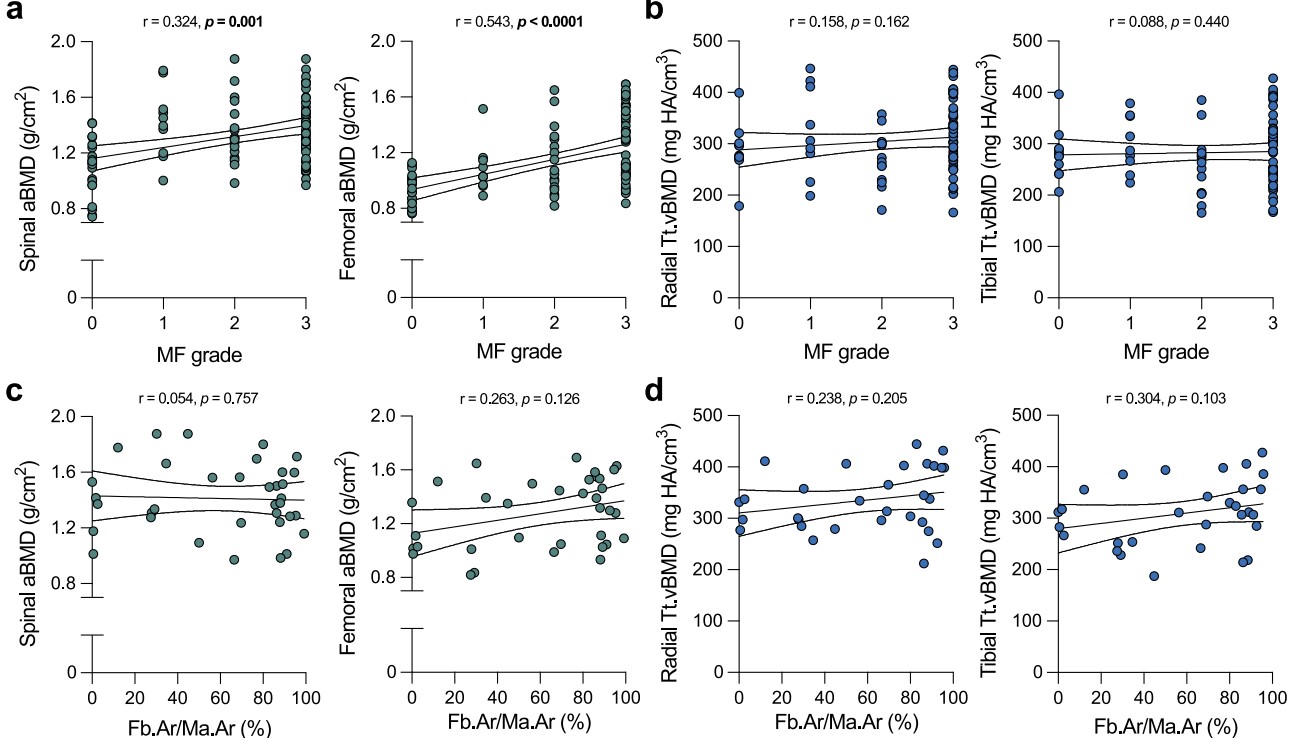

**Fig. 6 | Areal bone mineral density (aBMD) assessed by dual-energy X-ray absorptiometry (DXA) is associated with bone marrow fibrosis grade (MF grade). a** Association of MF grades with spinal and femoral aBMD assessed by DXA in myelofibrosis patients (n = 101). **b** Association of MF grades with radial and tibial total vBMD (Tt.vBMD) assessed by high-resolution peripheral quantitative computed tomography (HR-pQCT) in myelofibrosis patients (n = 80). **c** Association of fibrosis area to marrow area (Fb.Ar/Ma.Ar) with spinal and femoral aBMD (n = 35).

**d** Association of fibrosis area to marrow area (Fb.Ar/Ma.Ar) with radial and tibial Tt.vBMD (n = 30). All data points and simple linear regression analyses including 95% confidence intervals are shown. Associations between the presented parameters were calculated using Spearman's rank correlation (**a**–**d**). The correlation coefficient r with the corresponding exact *p*-value is shown above each graph. Numbers in bold indicate statistical significance (*p* < 0.05). Source data are provided as a Source Data file.

values, which may overestimate bone quality and HR-pQCT should be used to assess fracture risk in MF patients, which is also reflected by the high predictive value of HR-pQCT for fractures in general[30,31]. In conjunction with the stable HR-pQCT parameters following allo-HSCT, it is unlikely that the drastic reduction in DXA aBMD is directly linked to an increased fracture risk, but rather to normalized skeletal homeostasis. However, DXA should still be included in the long-term monitoring of fracture risk in MF patients after allo-HSCT, as, for instance, changes in aBMD at central sites were greatest in the context of osteoporosis medications such as romosozumab[32], and may therefore also indicate early trends in BMD reduction.

Clinical chemistry analysis for serum and urinary bone metabolism parameters showed a pattern of disturbed calcium homeostasis before transplantation, which was expressed by the fact that more than half of the patients had secondary hyperparathyroidism. These findings are in line with the observed poor bone mineralization in iliac crest biopsies, in terms of high osteoid levels and low mean matrix mineralization (CaMean) levels detected by qBEI. Together, this pattern suggests that the dynamic formation of new bone in MF (i.e., osteosclerosis) is dependent on a high demand for calcium. We demonstrated that calcium homeostasis was restored after allo-HSCT, along with the decline of osteosclerosis. Importantly, the bone remodeling capacity, responsible for the continuous renewal of the healthy skeleton, was also restored, as evidenced by the increase in bone turnover parameters. Although purely speculative, an inadequate normalization of serum parathyroid hormone or increase of urinary DPD in the early phase after allo-HSCT (30 days) could be indicative of a treatment failure with particularly high diagnostic value for the outcome osteosclerosis.

In conclusion, we present a comprehensive assessment of the trajectories of bone mass, microarchitecture, matrix quality, and bone turnover in MPN-associated MF patients undergoing allo-HSCT. We demonstrate that the formation of osteosclerosis is accompanied by a massive calcium demand associated with hyperparathyroidism. While allo-HSCT induces a rapid normalization of osteosclerosis, recovered osteoclast function along with restored whole organism calcium homeostasis, our findings show that high-resolution peripheral measurement of bone microarchitecture is the method of choice to adequately assess the skeletal status of patients with MPN-associated MF.

## Methods

### Ethics approval

Ethics approval was obtained from the local ethics committee (Ethikkommission der Ärztekammer Hamburg, 2021-100659-BO-ff) and all patients gave informed written consent. This study was conducted in accordance with the Declaration of Helsinki.

### Study cohort and propensity score matched healthy controls

A total of 66 patients were recruited for this prospective cohort study between May 2021 and August 2023. Patients with MPN-associated MF (PMF, post-ET MF, post-PV MF) presented to the Department of Stem Cell Transplantation, University Medical Center Hamburg-Eppendorf. Due to the transplantation setting, most patients included in this study showed late-stage disease progression and heavy fibrosis. Patients suffering from disease relapse, secondary acute leukemia, graft failure, or death after transplantation were excluded. As part of the routine diagnostic workup, disease-specific characteristics (DIPPS[20]/MYSEC-PM[21], MF grade[19]) and demographic characteristics were obtained (Supplementary Table 1). For allo-HSCT, patients were transplanted with the best available donor. This included matched related (MRD), haploidentical related (haploMMRD), matched unrelated (MUD), and

mismatched unrelated (MMUD) donors. Of the 66 patients included, 9 (13.6%) received a transplant from an MRD, while 47 (71.2%) received a transplant from a MUD. Eight patients (12.1%) had an MMUD available and two (3.0%) had a haploMMRD. All patients received an alkylating agent-based conditioning regimen. Of the 66 patients enrolled, 58 patients (87.9%) received a reduced-intensity conditioning dose containing busulfan and 8 patients (12.1%) received a myeloablative conditioning dose containing treosulfan. Patients with a MUD or MMUD received 60 mg/kg body weight antithymocyte globulin (ATG), while patients with a MRD received 30 mg/kg body weight ATG. Patients with haploMMRD did not receive ATG, but were switched to cyclophosphamide-based GvHD prophylaxis at 50 mg/kg body weight on days 3 and 4 after transplantation.

Densitometry, Jamshidi iliac crest biopsy, and clinical chemistry assessment were performed before and after allo-HSCT at standardized time points (i.e., 30 days, 100 days, 1 year). While the measurements could not be obtained at all time points in all patients, those patients with multiple measurements at different time points were analyzed separately. Serum parameters for clinical chemistry analyses were available for a larger cohort of up to 123 MF patients. When comparing patients who received advanced skeletal imaging with those who received only laboratory analysis, the two cohorts did not differ in terms of key clinical and laboratory characteristics.

A control cohort was retrospectively included and matched by propensity score accounting for sex, age, and BMI. These patients underwent skeletal assessment due to external referral because of suspected but not confirmed bone disease. A comprehensive skeletal assessment including DXA and HR-pQCT measurement on the same day was required for inclusion. Patients with medical conditions causing major effects on bone health were excluded: genetic bone diseases (e.g., osteogenesis imperfecta, osteopetrosis, and osteosclerosis), active cancer or tumors with potential systemic or local effects on BMD (e.g., myeloma and skeletal metastases), history of allo-HSCT, high-dose corticosteroid treatment (≥ 7.5 mg/day of prednisolone or equivalent), conditions or neurological disorders associated with prolonged generalized or local immobilization (e.g., Parkinson's disease, multiple sclerosis, and apoplexy), chronic kidney disease (glomerular filtration rate < 30 mL/min).

In-depth histological characterization was performed on a total of 60 iliac crest biopsies, including 20 specimens each for MF patients before and after allo-HSCT as well as controls. The iliac crest biopsies of the controls had been obtained postmortem in the context of a previous study[22], and the above-mentioned conditions affecting bone health were excluded during full autopsy. Selection of the specimens from a larger cohort was conducted by propensity score matching for sex, age, BMI, and 25-OH-D to allow comparability of the three groups.

### DXA and HR-pQCT

To determine the overall skeletal status, MF patients presented to the Department of Osteology and Biomechanics, University Medical Center Hamburg-Eppendorf. DXA (Lunar iDXA, GE Healthcare, Madison, WI, USA) was performed at the lumbar spine and hip. The areal BMD was evaluated and corresponding T-scores (i.e., the standard deviation compared to 20-30-year-old healthy, sex-matched individuals) and Z-scores (i.e., the standard deviation compared age- and sex-matched healthy individuals) were calculated by the software provided by the manufacturer. The aBMD, T-score, and Z-score of the lumbar spine (L1-L4) and the left hip (total hip) were utilized for further analyses. Daily calibration scans were performed with a dedicated phantom following the manufacturer's recommendations for DXA quality assurance. This includes precision testing including least significant change calculations according to the recommendations of the International Society for Clinical Densitometry[33].

Subsequently, HR-pQCT (XtremeCT, first generation, Scanco Medical, Switzerland) was performed to determine the vBMD,

microstructure, and geometry at the non-dominant distal radius and the contralateral distal tibia in a standardized procedure using the in vivo protocol (59.4 kVp, 900 µA, 100 ms integration time, 82.0 µm voxel size). The scan region begins at a fixed offset distance of 9.5 and 22.5 mm proximal to the reference line, which is placed at the inflection point of the endplate of the distal radius and tibia plafond, respectively. The total scan region extends 110 slices (9.02 mm) proximally from this point[34]. HR-pQCT images were reviewed for motion artifacts and excluded if motion artifacts grade 4 or 5 were detected[35]. Namely, we assessed the volumetric total, trabecular and cortical BMD (Tt.vBMD, Tb.vBMD, Ct.vBMD), the bone volume to tissue volume (BV/TV), trabecular number (Tb.N), thickness (Tb.Th), separation (Tb.Sp), cortical thickness (Ct.Th), and cortical perimeter (Ct.Pm) as well as total, trabecular and cortical area (Tt.Ar, Tb.Ar, Ct.Ar) according to current guidelines[34]. HR-pQCT results were compared with device-, age-, and sex-specific reference values[36].

### Undecalcified histology & histomorphometry

To assess the histological changes before and after allogeneic hematopoietic stem cell transplantation, standardized iliac crest biopsies were obtained. All biopsies were fixed at 4 °C in 3.5% PBS buffered formaldehyde. They were embedded undecalcified in methyl methacrylate, cut into 4 µm sections, and stained according to standard procedures (Gomori's trichrome, trichrome Masson-Goldner, von Kossa, and toluidine blue). Static histomorphometry was performed on the stained undecalcified histological bone sections. Histomorphometry analysis was carried out according to the nomenclature of the American Society for Bone and Mineral Research (ASBMR)[37] using an OsteoMeasure histomorphometry system (Osteometrics, Atlanta, GA, USA) connected to a Zeiss microscope (Carl Zeiss, Jena, Germany). The fibrosis area to marrow area (Fb.Ar/Ma.Ar) and the area of adipose tissue (Ad.Ar/Ma.Ar) were measured. Furthermore, the bone volume to tissue volume (BV/TV), mineralized volume to tissue volume (Md.V/TV), trabecular thickness (Tb.Th), trabecular number (Tb.N), trabecular separation (Tb.Sp), and the unmineralized (i.e., osteoid) volume per bone volume (OV/BV), osteoid surface per bone surface (OS/BS), and osteoid thickness (O.Th) were quantified. Osteomalacia was defined as OV/BV > 2%[22]. We also measured cellular bone turnover parameters, i.e., the number of bone-forming osteoblasts and bone-resorbing osteoclasts per bone perimeter (N.Ob/B.Pm, N.Oc/B.Pm) as well as their surface to bone surface (Ob.S/BS, Oc.S/BS).

### Quantitative backscattered electron imaging (qBEI)

Following histological sectioning, the embedded specimens were polished to a coplanar finish and carbon-coated. A scanning electron microscope (LEO 435 VP, LEO Electron Microscopy Ltd., Cambridge, England) with a backscattered electron detector (Type 202, K.E. Developments Ltd., Cambridge, England; 20 kV, 680 pA, 20 mm working distance) was utilized to obtain the bone mineral density distribution (BMDD) and quantify osteocyte lacunar parameters. In short, the mean calcium content (CaMean), mineralization heterogeneity (CaWidth), and the fraction of lowly and highly mineralized matrix (CaLow, CaHigh) were derived from the images using a custom MATLAB-based script (MathWorks, Natick, MA, USA). Cutoff levels for CaLow and CaHigh correspond to the 5th and 95th percentile of the control group, respectively. Furthermore, the number of osteocyte lacunae per bone area (N.Ot.Lc/B.Ar) and the mean osteocyte lacunar area (Ot.Lc.Ar) were assessed.

### Laboratory assessment

To assess mineral and bone metabolism, blood and urine samples were collected and analyzed in the Department of Clinical Chemistry, University Medical Center Hamburg-Eppendorf. Analyses included calcium, phosphate, 25-hydroxyvitamin D (25-OH-D), parathyroid hormone (PTH), alkaline phosphatase (ALP), bone-specific ALP, and

osteocalcin in the blood serum as well as deoxypyridinoline (DPD) per creatinine in the urine. Reference values were provided by the local laboratory. Vitamin D insufficiency was defined as 25-OH-D level < 30 ng/ml.

### Statistical analysis

Statistical analysis was performed using SPSS Statistics 29.0.2 (IBM, Armonk, NY, USA) and GraphPad Prism 10.4.1 (GraphPad Software, San Diego, CA, USA). Results are reported as mean ± standard deviation (SD) and with mean percentage of the median of reference values for HR-pQCT parameters. To allow comparability between the investigated study groups, we applied propensity score matching for selected parameters. Differences in frequency distributions were tested by Fisher's exact test and corresponding effect sizes were reported as $\omega$ ($0.1 \triangleq$ small, $0.3 \triangleq$ medium, $0.5 \triangleq$ large effect size). The Shapiro–Wilk test was used to evaluate normal distribution of the data. To test for differences between two independent groups, the unpaired two-tailed $t$ test was used for normally distributed data and the Mann–Whitney $U$ test was used for non-parametric data. To test for differences between two dependent subgroups, the paired two-tailed $t$ test was used for normally distributed data and the Wilcoxon signed-rank test was used for non-parametric data. Effect sizes for comparisons between two groups were reported as Cohen's d ($0.2 \triangleq$ small, $0.5 \triangleq$ medium, $0.8 \triangleq$ large effect size). When testing for differences between three groups, one-way analysis of variance (ANOVA) with Holm–Šidák test was used for normally distributed data and Kruskal–Wallis $H$ test with Dunn's test for non-parametric data, corresponding effect sizes were reported as $\eta^2$ ($0.02 \triangleq$ small, $0.13 \triangleq$ medium, $0.26 \triangleq$ large effect size). To evaluate the associations between bone marrow fibrosis and BMD, Spearman's rank correlations were performed and the correlation coefficients r with corresponding $p$-values were calculated. Simple linear regression analyses including 95% confidence intervals of the respective regression slopes were also presented. To predict changes in aBMD after allo-HSCT non-linear regression analyses including 95% confidence intervals and corresponding $p$-values with the coefficient of determination $R^2$ were calculated. The level of statistical significance was defined as $p < 0.05$.

### Reporting summary

Further information on research design is available in the Nature Portfolio Reporting Summary linked to this article.

## Data availability

All data supporting the results of this study are available in the article, Supplementary Information and/or Source Data file. Source data are provided with this paper.

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

## Acknowledgements

We thank Silke Zeschke for sample collection and preparation as well as Andrea Thieke and Maximilian Lenard Thiessen for excellent technical assistance in preparing the histological sections. This research did not receive any specific grant from funding agencies in the public, commercial, or not-for-profit sectors. We acknowledge financial support from the Open Access Publication Fund of UKE - Universitätsklinikum Hamburg-Eppendorf.

## Author contributions

M.A., N.K., and T.R. conceptualized the study. All authors (M.S., A.S., T.A.Y., F.N.v.B., M.M.D., A.B., N.G., F.A., T.S., M.A., N.K., and T.R.) designed the methodology and participated in patient recruitment, sample collection, and clinical data acquisition. M.S., A.S., T.A.Y., A.B., T.S., M.A., N.K., and T.R. investigated the samples and conducted the measurements. A.S. and T.R. carried out the formal analysis and visualized the data. All authors validated the data and contributed to the interpretation of the results. M.S., A.S., and T.R. wrote the original draft. All authors reviewed and edited the manuscript. The project was supervised by M.A., N.K., and T.R., with N.K., T.R. administering the project.

## Funding

## Competing interests

The authors declare no competing interests.
