## [Transparent Peer Review file · Nature Communications]

Reconstructing skeletal homeostasis through allogeneic hematopoietic stem cell transplantation in myelofibrosis

Corresponding Author: Dr Tim Rolvien

Version 0:

Reviewer comments:

Reviewer #1

(Remarks to the Author)

This is a nicely conducted study and a well-written paper, with an in-depth characterization of skeletal changes of myelofibrosis patients before and after allo-HSCT.

Methodology, results and validity

1. I have no fundamental comments to the methodology and statistic approach.

The results are important and novel and help understanding the skeletal changes in myelofibrosis.

Comments and suggestions

2. One of the notable clinical findings is that in myelofibrosis patients higher T-scores were found in the aBMD measurement of the spine and hip, compared to healthy controls and that after stem cell transplantation a clear change in the aBMD occurred while this was not the case with the HRPQCT measurement at the distal radius and tibia.

This is an interesting and important finding and the authors may elaborate a bit more on this discrepancy in the discussion.

On the one hand, this could be due to the fact that the HRPQCT measurement does not pick up changes as well as with DXA but, on the other hand, there could be a difference between changes in the central skeleton versus the peripheral skeleton as the authors already mentioned.

In my opinion, it seems unlikely that with HRPQCT no change in bone parameters could be observed merely due to technical issues (compared to DXA). It may also not be entirely ruled out that the changes measured with DXA could be at least partly related to a change in bone microarchitecture, especially since this study has shown a change in osteoclast function and calcium homeostasis before and after HSCT.

The question is whether, based on this study, it can be stated that DXA would not be suitable to assess fracture risk in patients suffering from MF, as stated on page 9. For example, in patients treated with Romosozumab or other osteoporosis medication, the changes in aBMD of central bone sites have been shown to be (much) greater than at the radius (table S6 in: Mc Clung et al. Romosozumab in Postmenopausal Women with Low Bone Mineral Density. N. Engl. J. Med. 370, 412–420 (2014)). So, it could also be that changes in peripheral bone may be less pronounced than in central bone (see reference NEJM) and that a peripheral bone measurement may be less representative for the fracture risk in these patients. In that context, it may also be important whether the risk of more central (hip, vertebra) or peripheral (radius, tibia) fractures is considered.

3. On page 9 in the discussion, it is mentioned that: "The high degree of MF in the histomorphometric analyses of iliac crest biopsies moderately correlated with elevated DXA T-scores in MF patients. Not only does this mean that standard DXA osteodensitometry may not be suitable to assess fracture risk in patients suffering from MF, but it might instead be suitable to assess the degree of fibrosis in these patients non-invasively."

It would be interesting to apply the Trabecular Bone Score analysis to the DXA images of the spine. Is it possible to add the TBS data to this study? This would be particularly interesting because it would allow for an additional examination of vertebral structural bone parameters and to study its association with the degree of MF.

Minor comments and suggestions

4. I would suggest that the authors change the terminology proximal and distal fractures (page 10) to central and peripheral.
5. Please add the generation of the HRpQCT scanner (probably the first generation HRpQCT device has been used)
6. Is it possible to add data on cortical porosity to the HRpQCT analyses?

Reviewer #2

(Remarks to the Author)

In this article Schäfersküpper & Simon et al. provide an in depth assessment of skeletal homeostasis following allogeneic stem cell transplantation in patients with myelofibrosis.

The article is well written und conducted. In general, bone health in patients following allo-Tx remains a topic where solid scientific data remains limited (many studies are either retrospective, with a small cohort size or lack control groups).

This study is conducted in a relatively large cohort (for this condition), with relevant control groups and conducted in a prospective manner.

Bone quality is assessed through a number of different methods including (DXA; pQCT, histological and biomarker analyses).

Notable results are:

- rapid and drastic reduction/ normalization of BMD at spine and hip in MF cohort, following allo-TX
- extensive restructuring of bone matrix
- evidence of restored osteoclastic resorption capacity after allo-Tx

The results drawn from this study are conclusive and of interest to the field. The authors have gone a long way to gather this depth of information from a relatively rare disease. The finding of a correlation between fibrosis and T-Score is an interesting finding that warrants further research with regard as a potential non-invasive prognostic marker and I would encourage the authors to follow-up on this idea.

Minor comments:

Figure 1: While nice to look at, I am not sure that the radiographs in Figure 1 really improve the manuscript. In fact, figure 1b,c,d and e are relatively small. If the authors decide to include the radiographs they should consider rearranging the figure to assure good readability of the diagrams.

Figure 2: same applies to figure 2.

Figure 5: is the pre-transplant time-point more clearly defined? If so, consider labelling -XXd

Reviewer #3

(Remarks to the Author)

In this study, the investigators performed comprehensive longitudinal evaluations of myelofibrosis (MF) patients before and after allogeneic stem cell transplant (allo-HSCT) to characterize the skeletal changes using clinical high-resolution imaging, laboratory analyses, and bone biopsy studies. They observed a strong elevation of bone mineral density at the lumbar spine and proximal femur, fully normalized after allo-HSCT. Importantly, the regression of fibrosis was accompanied by vanishing osteosclerosis along with restored osteoclastic resorption activity and calcium homeostasis.

This is a very interesting study providing evidence for an extensive reconstruction of skeletal homeostasis by allo-HSCT in MF, leading to rapid resolution of osteosclerosis. Such in depth mechanistic reporting of the skeletal changes in MF patients post allo-HSCT to my knowledge have not been previously reported and provide an important contribution to our field.

Major comments

1. The authors report that there were no differences in the high risk MF features between men and women, but what about such differences between the primary MF vs post-ET vs post-PV groups?
2. Similarly were there any differences in the bone healing in those three primary MF vs post-ET vs post-PV groups?
3. More detail on the allo-HSCT treatment would be helpful.
 - a. Did all 66 patients get the same preparative therapy?
 - b. Did the preparative therapy contain radiation?
 - c. Did the patients all receive matched donors or were mismatched donors included?
 - d. If the preparative regimen or type of donor were different, were there difference in bone healing in those subgroups?
4. A bone-healthy control cohort was retrospectively included and matched by propensity score accounting for sex, age, BMI, and 25-OH-D. These patients underwent skeletal assessment due to external referral because of suspected but not

confirmed bone disease. However they report that The iliac crest biopsies of the healthy controls had been acquired in the context of a previous study during autopsy. This needs to be clarified as obviously they would not be healthy if the bone marrow biopsies were obtained at autopsy

5. Having a control group of patients undergoing allo-HSCT for diseases other than MF would enhance enthusiasm for this paper. There are bone changes that occur due to chemo/steroids/reduced mobility post-transplant that could helped solidify their conclusions

6. The authors report on the osteoclastic resorption and calcium homeostasis on a larger cohort of 123 MF patients.

a. What were the characteristic of these additional 57 patients and why weren't the included in the initial description?

b. Were the results similar to the 66 patients included in the original cohort?

7. The authors conclude that while allo-HSCT induces a rapid normalization of osteosclerosis, recovered osteoclast function along with restored whole organism calcium homeostasis, their findings show that high-resolution peripheral measurement of bone microarchitecture is the method of choice to adequately assess the skeletal status of patients with MPN-associated MF.

a. Are their clinical implications to this finding ie recommending that test to follow MF patients post allo-HSCT

b. Can the authors hypothesize if doing some of these tests earlier than one year post-allo-HSCT might allow for other therapeutic interventions to increase the rate of bone healing?

Version 1:

Reviewer comments:

Reviewer #1

(Remarks to the Author)

The authors have addressed my comments adequately, I have no further comments.

Reviewer #2

(Remarks to the Author)

The authors have addressed all relevant comments made by the reviewers, which has further improved the manuscript.

Reviewer #3

(Remarks to the Author)

I reviewed the revised paper and believe the authors have addressed all the reviewers issues satisfactorily i have no other suggestions

Point-by-point response to the reviewers

Response to reviewer #1:

This is a nicely conducted study and a well-written paper, with an in-depth characterization of skeletal changes of myelofibrosis patients before and after allo-HSCT.

Response: We would like to thank the reviewer for the positive feedback on our manuscript and the suggestions made. Detailed answers to the questions and comments can be found below.

Methodology, results and validity

1. I have no fundamental comments to the methodology and statistic approach.

The results are important and novel and help understanding the skeletal changes in myelofibrosis.

Response: We are delighted that the reviewer finds our analyses technically sound and the findings original.

Comments and suggestions

2. One of the notable clinical findings is that in myelofibrosis patients higher T-scores were found in the aBMD measurement of the spine and hip, compared to healthy controls and that after stem cell transplantation a clear change in the aBMD occurred while this was not the case with the HRpQCT measurement at the distal radius and tibia.

This is an interesting and important finding and the authors may elaborate a bit more on this discrepancy in the discussion.

On the one hand, this could be due to the fact that the HRpQCT measurement does not pick up changes as well as with DXA but, on the other hand, there could be a difference between changes in the central skeleton versus the peripheral skeleton as the authors already mentioned. In my opinion, it seems unlikely that with HRpQCT no change in bone parameters could be observed merely due to technical issues (compared to DXA). It may also not be entirely ruled out that the changes measured with DXA could be at least partly related to a change in bone microarchitecture, especially since this study has shown a change in osteoclast function and calcium homeostasis before and after HSCT.

*The question is whether, based on this study, it can be stated that DXA would not be suitable to assess fracture risk in patients suffering from MF, as stated on page 9. For example, in patients treated with Romosozumab or other osteoporosis medication, the changes in aBMD of central bone sites have been shown to be (much) greater than at the radius (table S6 in: Mc Clung et al. Romosozumab in Postmenopausal Women with Low Bone Mineral Density. *N. Engl. J. Med.* 370, 412–420 (2014)). So, it could also be that changes in peripheral bone may be less pronounced than in central bone (see reference *NEJM*) and that a peripheral bone measurement may be less representative for the fracture risk in these patients. In that context, it may also be important whether the risk of more central (hip, vertebra) or peripheral (radius, tibia) fractures is considered.*

Response: We thank the reviewer for this thoughtful comment, which allowed us to elaborate more on the discrepancy between central aBMD (DXA) and peripheral 3D microstructural data (HR-pQCT) in the discussion. We also believe that less technical reasons but rather the fact of deviating measurement regions is the main reason for the considerable discrepancies between increased aBMD (DXA) but normal vBMD and microarchitecture (HR-pQCT). Fitting to this assumption is that our histomorphometric analysis in the iliac crest showed not only a higher degree of marrow fibrosis but also osteosclerosis, i.e. increased T-scores in the nearby measurement regions (lumbar spine, femur) may indeed be due to increased bone mass. In addition, in a previous study, some of the authors were able to show that metabolic activity in MF patients progresses from central to peripheral bone marrow areas, as detected by ¹⁸F-FDG PET/CT28 (Derlin, T., et al., Kröger, N. *J Nucl Med.* 57, 1556-1559 (2016)). Together, our findings suggest that DXA measurements indicate elevated aBMD values, which may overestimate

bone quality and HR-pQCT should be used to assess fracture risk in MF patients, which is also reflected by the high predictive value of HR-pQCT for fractures in general (Sarfati, M., et al., Bouxsein, M.L. *J Bone Miner Res.* 39, 1574-1583 (2024)). In conjunction with the stable HR-pQCT parameters following allo-HSCT, it is unlikely that the drastic reduction in DXA aBMD is directly linked to an increased fracture risk, but rather to normalized skeletal homeostasis. However, as suggested by the reviewer, we acknowledge that DXA should be included in the long-term monitoring of fracture risk after allo-HSCT, as changes in aBMD at central sites were greatest in the context of osteoporosis medications such as romosozumab (McClung, M.R., et al., Bone, H.G. *N Engl J Med.* 370, 412-420 (2014)) and may therefore also indicate early trends in BMD reduction. These points have now been included in the discussion.

3. *On page 9 in the discussion, it is mentioned that: "The high degree of MF in the histomorphometric analyses of iliac crest biopsies moderately correlated with elevated DXA T-scores in MF patients. Not only does this mean that standard DXA osteodensitometry may not be suitable to assess fracture risk in patients suffering from MF, but it might instead be suitable to assess the degree of fibrosis in these patients non-invasively."*

It would be interesting to apply the Trabecular Bone Score analysis to the DXA images of the spine. Is it possible to add the TBS data to this study? This would be particularly interesting because it would allow for an additional examination of vertebral structural bone parameters and to study its association with the degree of MF.

Response: We regret that the TBS software is not available at our institution. We understand that the application of TBS to the lumbar spine DXA scans would be particularly interesting because of the central measurement region, which cannot be measured by HR-pQCT. While indeed interesting, we believe that TBS is of primary interest to clinicians who do not have HR-pQCT to get some idea into bone microarchitecture. We hope that, together with the detailed biopsy analyses, we still provide enough data to support a reasonable explanation for the cause of the elevated DXA values (i.e., most likely due to osteosclerosis, i.e., true increase in bone mass).

Minor comments and suggestions

4. *I would suggest that the authors change the terminology proximal and distal fractures (page 10) to central and peripheral.*

Response: Thank you, the terms have now been uniformly changed to "central" and "peripheral".

5. *Please add the generation of the HRpQCT scanner (probably the first generation HRpQCT device has been used)*

Response: The first-generation HR-pQCT scanner was used. This information was now added to the Methods section.

6. *Is it possible to add data on cortical porosity to the HRpQCT analyses?*

Response: Indeed, it is theoretically possible to calculate cortical porosity from HR-pQCT images. However, our experience with the daily use of both first- and second-generation HR-pQCT scanners has shown that this parameter is inaccurate when applied to the first-generation scanner. Technically, the measurement of Ct.Po is limited by the spatial resolution of the HR-pQCT images (134.6-154.4 μm spatial resolution for first-generation HR-pQCT), as the size of the Haversian canals typically ranges from 30 to 350 μm (Whittier, D.E., et al., Bouxsein, M.L. *Osteoporos Int.* 31, 1607-1627 (2020)). An assessment of cortical porosity with the standard first-generation cortical analysis should therefore be performed with caution, which is why we decided not to report this parameter.

Response to reviewer #2:

In this article Schäfersküpfer & Simon et al. provide an in depth assessment of skeletal homeostasis following allogeneic stem cell transplantation in patients with myelofibrosis. The article is well written and conducted. In general, bone health in patients following allo-Tx remains a topic where solid scientific data remains limited (many studies are either retrospective, with a small cohort size or lack control groups). This study is conducted in a relatively large cohort (for this condition), with relevant control groups and conducted in a prospective manner. Bone quality is assessed through a number of different methods including (DXA; pQCT, histological and biomarker analyses).

Notable results are:

- rapid and drastic reduction/ normalization of BMD at spine and hip in MF cohort, following allo-TX*
- extensive restructuring of bone matrix*
- evidence of restored osteoclastic resorption capacity after allo-Tx*

The results drawn from this study are conclusive and of interest to the field. The authors have gone a long way to gather this depth of information from a relatively rare disease. The finding of a correlation between fibrosis and T-Score is an interesting finding that warrants further research with regard as a potential non-invasive prognostic marker and I would encourage the authors to follow-up on this idea.

Response: We would like to thank the reviewer for the positive feedback on our study. Indeed, we have made great efforts to obtain a comprehensive understanding of the skeletal status in myelofibrosis and its changes after allo-HSCT. In the future, we plan to further investigate the role of DXA in the prediction of myelofibrosis grade before transplantation as well as regression after allo-HSCT. We will also continue to investigate the predictive power of certain laboratory parameters such as parathyroid hormone or DPD, which can be measured early after allo-HSCT. Please find detailed responses to the points mentioned below.

Minor comments:

Figure 1: While nice to look at, I am not sure that the radiographs in Figure 1 really improve the manuscript. In fact, Figure 1b,c,d and e are relatively small. If the authors decide to include the radiographs they should consider rearranging the figure to assure good readability of the diagrams.

Figure 2: same applies to figure 2.

Response: In line with the reviewer's comment, we have enlarged the labelling of the diagrams to improve readability. We have also uploaded the figures again in high quality. We would like to keep the radiological images (DXA and HR-pQCT) in Fig. 1a and Fig. 2a to give readers who are not from the bone field an insight into what was actually measured. The image details have been slightly adjusted to recognize the relevant aspects. For instance, the osteosclerotic changes (Fig. 1a), which resolved after allo-HSCT, can also be recognized.

Figure 5: is the pre-transplant time-point more clearly defined? If so, consider labelling -XXd

Response: The conditioning treatment started on day 6 before transplantation and all patients were examined before chemotherapy was administered. However, the timing varied from a few days to four weeks before the start of conditioning therapy. Since bone marrow changes are already well advanced in myelofibrosis patients who are eligible for transplantation, we do not expect large differences in bone measurements in such a small time frame. As no exact day could be defined, we would like to keep the term 'pre'.

Response to reviewer #3:

In this study, the investigators performed comprehensive longitudinal evaluations of myelofibrosis (MF) patients before and after allogeneic stem cell transplant (allo-HSCT) to characterize the skeletal changes using clinical high-resolution imaging, laboratory analyses, and bone biopsy studies. They observed a strong elevation of bone mineral density at the lumbar spine and proximal femur, fully normalized after allo-HSCT. Importantly, the regression of fibrosis was accompanied by vanishing osteosclerosis along with restored osteoclastic resorption activity and calcium homeostasis.

This is a very interesting study providing evidence for an extensive reconstruction of skeletal homeostasis by allo-HSCT in MF, leading to rapid resolution of osteosclerosis. Such in-depth mechanistic reporting of the skeletal changes in MF patients post allo-HSCT to my knowledge have not been previously reported and provide an important contribution to our field.

Response: We are grateful for the recognition of our work and hope that with the changes made, the manuscript is now fully acceptable for publication in Nature Communications.

Major comments

1. *The authors report that there were no differences in the high risk MF features between men and women, but what about such differences between the primary MF vs post-ET vs post-PV groups?*

Response: We thank the reviewer for this important comment. Risk stratification was performed using the DIPSS for patients with primary myelofibrosis, while the MYSEC was used for secondary myelofibrosis (post-ET and post-PV). Although the elements used to calculate the risk groups are similar between these two scoring systems, there is a small difference in median overall survival, which is why we did not make a direct comparison. All patients eligible for transplantation had either advanced stage disease or other molecular high-risk factors. However, according to the reviewer's suggestion, we did make a comparison between primary and secondary myelofibrosis, which can be found in the table for the reviewer below. There was a slight difference in the frequency of intermediate-2 vs. intermediate-1 risk, which we consider to be of minor importance due to the transplantation setting.

Table for the reviewer 1: Clinical characteristics of the study cohort prior to allo-HSCT stratified by disease entity.

Parameter	Total n = 66	PMF n = 40	SMF n = 26	P	ω
Sex					
Women	34 (51.5%)	17 (42.5%)	17 (65.4%)	0.083	0.22
Men	32 (48.5%)	23 (57.5%)	9 (34.6%)		
MF grade					
MF-3	48 (72.7%)	29 (72.5%)	19 (73.1%)	0.580	0.14
MF-2	13 (19.7%)	9 (22.5%)	4 (15.4%)		
MF-1	5 (7.6%)	2 (5.0%)	3 (11.5%)		
MF-0	0 (0.0%)	0 (0.0%)	0 (0.0%)		
DIPPS/MYSEC-PM					
High risk	14 (21.2%)	9 (22.5%)	5 (19.2%)	0.026	0.35
Intermediate-2 risk	40 (60.6%)	28 (70.0%)	12 (46.2%)		
Intermediate-1 risk	11 (16.7%)	3 (7.5%)	8 (30.8%)		
Low risk	1 (1.5%)	0 (0.0%)	1 (3.8%)		

Abbreviations: Allo-HSCT, Allogeneic hematopoietic stem cell transplantation. DIPPS, Dynamic international prognostic scoring system for myelofibrosis. MF grade, Bone marrow fibrosis grade. MYSEC-PM, Myelofibrosis secondary to PV and ET prognostic model. PMF, Primary myelofibrosis. SMF, Secondary myelofibrosis. Absolute values are displayed with corresponding percentages. Differences in the frequency distribution between PMF and SMF were tested using Fisher's exact test, *P* values with corresponding effect sizes ω (0.1 \triangleq small, 0.3 \triangleq medium, 0.5 \triangleq large effect size) are presented.

2. Similarly were there any differences in the bone healing in those three primary MF vs post-ET vs post-PV groups?

Response: When comparing the DXA trajectories between patients with primary and secondary myelofibrosis, it is apparent that those with secondary myelofibrosis had a greater decrease in aBMD (a potential surrogate parameter for myelofibrosis and osteosclerosis) after allo-HSCT, which was not significant in primary myelofibrosis. However, the subgroups are rather small, and we are concerned that this difference is more likely due to insufficient statistical power. We have, however, integrated this comparison into the manuscript as a new Supplementary Fig. 2. Overall, we observed that the effect of BMD reduction was stronger in the proximal femur than in the lumbar spine.

Supplementary Fig. 2. T-score assessed by dual-energy X-ray absorptiometry (DXA) in primary and secondary myelofibrosis (MF) before and after allogeneic hematopoietic stem cell transplantation (allo-HSCT). a Spinal and femoral T-score before and after allo-HSCT in patients with primary MF. **b** Spinal and femoral T-score before and after allo-HSCT in patients with secondary MF.

3. More detail on the allo-HSCT treatment would be helpful.

a. Did all 66 patients get the same preparative therapy?

Response: We thank the reviewer for giving us the opportunity to refine the details of the allo-HSCT treatment. All patients received an alkylating agent-based conditioning regimen. Of the 66 patients enrolled, 58 (87.9%) received a reduced intensity conditioning (RIC) dose containing busulfan and 8 (12.1%) received a myeloablative conditioning (MAC) dose containing treosulfan. Patients with a matched unrelated donor (MUD) or mismatched unrelated donor (MMUD) received 60 mg/kg body weight antithymocyte globulin (ATG), while patients with a matched related donor (MRD) received 30 mg/kg body weight ATG. Patients with haploidentical related donors (haploMMRD) did not receive ATG, but were switched to cyclophosphamide-based GvHD prophylaxis at 50 mg/kg body weight on days 3 and 4 after transplantation. We have added details of the conditioning regimen and donor type to the Supplementary Table 1.

b. Did the preparative therapy contain radiation?

Response: No, the patients received an alkylating agent-based conditioning regimen and total body radiation was not used.

c. Did the patients all receive matched donors or were mismatched donors included?

Response: Patients were transplanted with the best available donor. This included matched related (MRD), haploidentical related (haploMMRD), matched unrelated (MUD), and mismatched unrelated (MMUD) donors. Of the 66 patients included, 9 (13.6%) received a transplant from an MRD, while 47 (71.2%) received a transplant from a MUD. 8 patients (12.1%) had an MMUD available and 2 (3.0%) had a haploMMRD. This information is now included in the Supplementary Table 1.

d. If the preparative regimen or type of donor were different, were there differences in bone healing in those subgroups?

Response: Due to the similarity of the preparative regimens, we do not find it useful to test for differences in bone healing with regard to conditioning therapy or donor selection. In particular, when differentiating between matched (58) and mismatched (8) donors, the latter group appears to be too small to derive meaningful results.

4. A bone-healthy control cohort was retrospectively included and matched by propensity score accounting for sex, age, BMI, and 25-OH-D. These patients underwent skeletal assessment due to external referral because of suspected but not confirmed bone disease. However they report that The iliac crest biopsies of the healthy controls had been acquired in the context of a previous study during autopsy. This needs to be clarified as obviously they would not be healthy if the bone marrow biopsies were obtained at autopsy.

Response: We thank the reviewer for the important comment and the opportunity to clarify. Indeed, the control group in the histologic studies was different from that of the clinical analysis. Specifically, we used a valuable database of 675 iliac crest biopsies (Priemel, M., et al., Amling, M. *J Bone Miner Res.* 25, 305-312 (2010)), which were obtained postmortem. A variety of conditions potentially affecting bone health as well as skeletal disorders were excluded in these individuals during full autopsy. From this database, we assembled the control group by propensity score matching to control for important factors such as age, sex and serum vitamin D (25-OH-D). All analyses such as histomorphometry and qBEI were performed on the iliac crest biopsies only for this study.

5. Having a control group of patients undergoing allo-HSCT for diseases other than MF would enhance enthusiasm for this paper. There are bone changes that occur due to chemo/steroids/reduced mobility post-transplant that could help solidify their conclusions

Response: We are very sorry, but this analysis is beyond the scope of the present study, in which we have made great efforts to investigate a prospective study of skeletal status in myelofibrosis. We also lack ethics approval for the inclusion of patients with other diseases. Nevertheless, the reviewer's comment is very important and we have therefore added and discussed additional literature in the revised manuscript to infer potential mechanisms. An indirect indication that the decrease in DXA values after allo-HSCT is due to resorbed pathologic bone (i.e., osteosclerosis) rather than actual bone loss in the context of established risk factors such as chemotherapy is the magnitude of BMD reduction compared to other malignancies of the hematopoietic system. Namely, the reduction in aBMD in our cohort, calculated one year after allo-HSCT, was -8.0% at the lumbar spine and -18.4% at the proximal femur (Supplementary Fig. 1a, b), compared to -5.5% at the lumbar spine and -6.8% at the proximal femur in patients with acute lymphoid leukemia (ALL), acute myeloid leukemia (AML), and myeloid dysplastic syndrome (MDS) (Lim, Y., et al., Kang, M.-I. *Bone.* 124, 40-46 (2019)).

6. The authors report on the osteoclastic resorption and calcium homeostasis on a larger cohort of 123 MF patients.

a. What were the characteristic of these additional 57 patients and why weren't they included in the initial description?

Response: Indeed, the full clinical data set, which included advanced clinical skeletal imaging, was available for 66 patients. During the course of the study, we were able to obtain laboratory trajectories focusing on bone remodeling and mineral metabolism markers in 57 additional patients who underwent allo-HSCT. The reasons why these patients did not undergo advanced skeletal imaging are manifold and mainly include lack of consent and insufficient physical fitness to undergo a several-hour examination in an outpatient clinic several kilometers away from the main clinical facility. We recognize this weakness and have nevertheless decided to report the laboratory data for a larger cohort.

b. Were the results similar to the 66 patients included in the original cohort?

Response: To address the potential limitation of two cohorts and possible bias associated with potential clinical differences, we compared patients who received advanced skeletal imaging (MF+) and those who received laboratory analysis only (MF). We found that the two cohorts did not differ with respect to key clinical features. Please find the Table for the reviewer 2 below. A respective explanation has also been added to the revised manuscript. The same applies for the comparison of laboratory parameters (Figure for the reviewer 1).

Table for the reviewer 2. Comparison of the clinical characteristics between the myelofibrosis patients who received advanced bone imaging (MF+, n=66) and myelofibrosis patients who did not receive advanced bone imaging (MF, n=57) before allo-HSCT.

Parameter	MF+ (n = 66)		MF (n = 57)		P	ω d
	Mean	SD	Mean	SD		
Demographics						
Sex (female / male)	34/32 (51.5%/48.5%)		20/37 (35.1%/64.9%)		0.072	0.17
Age (years)	60.3	8.7	61.4	7.7	0.352	0.01
Diagnosis						
PMF	40 (60.6%)		32 (56.1%)		0.822	0.06
Post-ET MF	16 (24.2%)		14 (24.6%)			
Post-PV MF	10 (15.2%)		11 (19.3%)			
DIPPS/MYSEC-PM						
High risk	14 (21.2%)		18 (31.6%)		0.395	0.16
Intermediate-2 risk	40 (60.6%)		33 (57.9%)			
Intermediate-1 risk	11 (16.7%)		6 (10.5%)			
Low risk	1 (1.5%)		0 (0.0%)			

Abbreviations: Allo-HSCT, Allogeneic hematopoietic stem cell transplantation. DIPPS, Dynamic international prognostic scoring system for myelofibrosis. MYSEC-PM, Myelofibrosis secondary to PV and ET prognostic model. PMF, Primary myelofibrosis. Post-ET MF, Post-essential thrombocythemia myelofibrosis. Post-PV MF, Post-polycythemia vera myelofibrosis.

Figure for the reviewer 1. Comparison of the laboratory analyses between the myelofibrosis patients who received advanced bone imaging (MF+, n=66) and myelofibrosis patients who did not receive advanced bone imaging (MF, n=57) before allo-HSCT. Differences between the groups were calculated using unpaired two-tailed *t* test for normally distributed data and the Mann–Whitney *U* test for non-parametric data. *P* values of the comparisons are displayed above the brackets.

7. The authors conclude that while allo-HSCT induces a rapid normalization of osteosclerosis, recovered osteoclast function along with restored whole organism calcium homeostasis, their findings show that high-resolution peripheral measurement of bone microarchitecture is the method of choice to adequately assess the skeletal status of patients with MPN-associated MF. a. Are there clinical implications to this finding ie recommending that test to follow MF patients post allo-HSCT

Response: We would like to thank the reviewer for the helpful comment. We have now further elaborated our recommendations on the clinical implications for fracture risk assessment in the revised manuscript. Accurate fracture risk assessment is important because patients suffering from myeloproliferative neoplasms are at higher risk of osteoporotic fractures. The generally advanced age of MF patients additionally contributes to an increased fracture risk, which is likely to increase further with the long-term survival achieved through allo-HSCT. Our results suggest that DXA measurements indicate elevated aBMD values, which may overestimate bone quality and HR-pQCT should be used to assess fracture risk in MF patients, which is also reflected by the high predictive value of HR-pQCT for fractures in

general (Sarfati, M., et al., Bouxsein, M.L. *J Bone Miner Res.* 39, 1574-1583 (2024)). In conjunction with the stable HR-pQCT parameters following allo-HSCT, it is unlikely that the drastic reduction in DXA aBMD is directly linked to an increased fracture risk, but rather to normalized skeletal homeostasis. However, DXA should still be included in the long-term monitoring of fracture risk in MF patients after allo-HSCT, as, for instance, changes in aBMD at central sites were greatest in the context of osteoporosis medications such as romosozumab (McClung, M.R., et al., Bone, H.G. *N Engl J Med.* 370, 412-420 (2014)) and may therefore also indicate early trends in BMD reduction.

b. Can the authors hypothesize if doing some of these tests earlier than one year post-allo-HSCT might allow for other therapeutic interventions to increase the rate of bone healing?

Response: Although purely speculative, an inadequate normalization of serum parathyroid hormone or increase of urinary DPD in the early phase after allo-HSCT (30 days) could be indicative of a treatment failure with particularly high diagnostic value for the outcome osteosclerosis.